# Integrin Alpha v Beta 6 (αvβ6) and Its Implications in Cancer Treatment

**DOI:** 10.3390/ijms232012346

**Published:** 2022-10-15

**Authors:** Ewa Brzozowska, Sameer Deshmukh

**Affiliations:** 1Hirszfeld Institute of Immunology and Experimental Therapy, Polish Academy of Sciences, St. R. Weigl 12, 53-114 Wroclaw, Poland; 2Research and Development Department, Pure Biologics Ltd., 54-427 Wroclaw, Poland

**Keywords:** integrin, αVβ6, cancer treatment, integrin beta 6 (INGB6)

## Abstract

Integrins are necessary for cell adhesion, migration, and positioning. Essential for inducing signalling events for cell survival, proliferation, and differentiation, they also trigger a variety of signal transduction pathways involved in mediating invasion, metastasis, and squamous-cell carcinoma. Several recent studies have demonstrated that the up- and down-regulation of the expression of αv and other integrins can be a potent marker of malignant diseases and patient prognosis. This review focuses on an arginine-glycine-aspartic acid (RGD)-dependent integrin αVβ6, its biology, and its role in healthy humans. We examine the implications of αVβ6 in cancer progression and the promotion of epithelial-mesenchymal transition (EMT) by contributing to the activation of transforming growth factor beta TGF-β. Although αvβ6 is crucial for proper function in healthy people, it has also been validated as a target for cancer treatment. This review briefly considers aspects of targeting αVβ6 in the clinic via different therapeutic modalities.

## 1. Introduction

Integrins are the proteins that function as cell-surface receptors and perform bidirectional signalling between the cells and their surroundings, thereby mediating cell adhesion, which supports cellular movements through adhesion and traction. These proteins are important in the regulation of various biological signalling pathways that control vital cellular processes such as cell survival, migration, proliferation, differentiation, metastasis, and tumour invasion. Two distinct α and β subunits form integrin heterodimers by non-covalent bonding. All the mammalian cells contain eighteen different α subunits and eight β subunits, which combine in numerous ways to form 24 kinds of integrin protein heterodimers with distinctive ligand-recognition and tissue-specific profiles. Several linked domains compose both integrin α and β subunits. Each subunit is structured from a transmembrane helix as well as a small intracellular cytoplasmic tail. The sizes of the subunits vary, but overall, α-subunits are usually composed of 1000 amino acids and β-subunits composed of 750 amino acids [1,2]. The integrin αvβ6 is unique from other heterodimeric proteins regarding the β subunit and was first reported in the 1990s. It was found to be poorly expressed or completely absent in the healthy adult tissue epithelia, whereas it was up-regulated specifically during tissue repair, carcinogenesis, and embryogenesis [3,4]. This review explores the αvβ6 integrin biology and its role in healthy humans. We examine the implication of this protein in cancer and its progression. Although αvβ6 is crucial for proper function in healthy people, it has gained momentum in cancer treatment as a potential target of interest. This review briefly considers aspects of targeting αVβ6 in the clinic via different therapeutic modalities.

## 2. Biology of the αvβ6 Integrin

### 2.1. Structure

The integrins consist of alpha and beta subtypes, which form transmembrane heterodimers. The subunit αv can bind to any of the β1, β3, β5, β6, or β8 subunit, but β6 is highly specific and only partners to the αv subunit. Three domains comprise both subunits: the endocellular, transmembrane, and extracellular domains. The β6 subunit has three regions: Cyto1, Cyto2, and Cyto3, all of which have a highly conserved amino acids sequence and are essential in establishing linkages to different cellular and extracellular signalling protein molecules [2]. Two types of integrin subunits span coding genes on different parts of human chromosomes but stay close to each other. The αv subunit is encoded by the gene that is also known as CD51 and located at 2q31-q32, whereas the integrin β6 subunit is encoded by the beta gene (ITGB6) residing on 2q24-q31 [5,6].

### 2.2. Ligands

Integrin αvβ6 contains tripeptide RGD (Arg-Gly-Asp) motif, which is highly specialized for adhesion to extracellular ligands. Both subunits collectively recognize the ligands that normally bind to the interface between the extracellular domains of both subunits (Figure 1).

There are sequences in the flanking areas of these subunits which determine the ligand specificity of the integrin receptor [1]. The cytoplasmic tail of the β6 subunit is confined to the adhesion of the receptor protein to the cell cytoskeleton, as well as to the intracellular signalling molecular networks [8]. This tail has an extension of 11 amino acids at its C-terminal, which makes it a characteristic tail not found in any other integrin protein, and contributes to the integrin αvβ6′ functions, especially those related to cancer [9,10]. Several molecular-level interactions have been reported modulating the integrin αvβ6 receptor functions.

Ligands that are specific to integrin αvβ6 include extracellular matrix molecules vitronectin, tenascin-C, and fibronectin, which are involved in cell adhesion and migration in vitro [11]. Latency-associated peptides (LAPs), which are the amino-terminal domains of transforming growth factors Beta (TGF-β) precursor peptides, form a latent TGF-β complex and act as ligands for the integrin αvβ6. When a binding of the integrin αvβ6 occurs with the latent TGF-β complex at the extracellular site via LAPs and at the same time with the intracellular cell cytoskeleton, the active TGF-β is released from the latent complex (Figure 2B,C). This activation is a significant physiological role of integrin αvβ6 in vivo [12].

### 2.3. Expression

Integrin αvβ6 is found to express in undetectable amounts in different situations, but it is considerably up-regulated particularly during tumorigenesis in various cancers where its expression is linked with poor prognosis. Research scientists have pointed out that αvβ6 is detected in high levels in epithelial cancers and is directly associated with tumour progression. The combined expression of αvβ6 and MMP-9 (Matrix metallopeptidase 9) proved to be an effective prognostic indicator in patients suffering from gastric cancer [14]. In the case of colorectal cancer, the combination of αvβ6 with eIF4E (eukaryotic translation initiation factor 4E) or Ets-1 was found to determine an effective prognosis [15,16]. High expression levels of integrin αvβ6 have been detected in patients with non-small cell lung cancer (NSCLC) and SCLC, and confirmed to correlate with positive patient survival [17,18]. On the other hand, a different research study including 215 NSCLC patients showed no association of αvβ6 expression with patient outcomes [19]. Similarly, any link between αvβ6 and survival was not evidenced in patients with ovarian cancer. However, the expression levels of integrin αvβ6 are positively correlated with pathological grades [20].

### 2.4. Regulation of Expression

The subunit β6 has the rate-limiting role in the formation of integrin αvβ6 heterodimer, as it can only dimerize with the αv subunit to form the receptor complex. Comparatively, the αv subunit shows high promiscuousness for other complex-forming subunits [1,2]. Therefore, the transcriptional activity of the β6 gene (ITGB6), which is normally high in epithelial cells, determines the expression levels of integrin αvβ6. Several binding sites are present on the ITGB6 promoter for the specific binding of transcription factors (Ets-1, signal transducer and activator of transcription 3 (STAT3), C/EBPα, SMAD3) [21]. ITGB6 expression in the normal epithelial cells is regulated by the SMAD3 activity, which is mediated by TGF-β1 [22], whereas in the transformed epithelial cells, other transcription factors (Ets-1, C/EBPα and STAT3) play a crucial role in its expression [21,23,24].

## 3. Crosstalk with Growth Factors in General and Specifically with TGF-β

The TGF-β family is highly conserved among different species and performs a variety of biological functions [25]. There are three types, TGF-β1, TGF-β2, and TGF-β3, in mammals. Each is synthesized in the form of a pro-TGF-β homodimer and has LAP covalently attached to it. Hence, each TGF-β type is secreted as a large latent TGF-β complex, which is formed by the disulphide covalent bonding between the latent TGF-β binding proteins (LTBPs) and small latent complex [26,27]. Different research investigations revealed that the integrin αvβ6 can activate both TGF-β1 and TGF-β3, whereas it cannot activate TGF-β2 [28].

In recent years, much interest has been shown towards the involvement of αvβ6-mediated TGF-β1 activation in tumorigenesis. TGF-β1 negatively regulates the proliferation in epithelial cells, protecting against environmental carcinogens. This prevention is carried out through up-regulating the cyclin dependent kinase inhibitors p15 and p21. Consequently, expression of the c-Myc, a promoter of cell proliferation, is suppressed [29]. Thus, epithelial cell growth is limited after the TGF-β1 activation and their quiescence is maintained. In the absence of αvβ6, epithelial hyperplasia may occur [30]. Contrary to the TGF-β1 activation by integrin αvβ6, TGF-β1 also maintains the expressions of the ITGB6 gene in ordinary epithelial cells. It shows mutual regulation between the two protein molecules [31]. The disturbance in the integrin αvβ6-TGF-β1 reciprocal regulatory mechanism leads to various αvβ6-associated pathological consequences [13].

During the investigation of gastric carcinoma, integrin αvβ6-positive cells were identified to carry vascular endothelial growth factor (VEGF). In the presence of VEGF, gastric carcinoma cells show cancer cell migration and an elevated expression of integrin αvβ6, and the extracellular signal-related kinase (ERK) pathway is activated. VEGF is also involved in the angiogenesis and invasion of carcinoma cells [32].

## 4. Implication of INGB6 in Cancer and Its Progression

There is an increasing interest in αvβ6-associated cancers due to high levels of αvβ6 expressions detected in many kinds of epithelial cancers, such as carcinomas of the skin, stomach, colon, breast, lung, oral mucosa, cervix, salivary gland, liver, ovary, and endometrium. The high expression of αvβ6 is mostly affiliated with metastasis and tumour invasion, and with a remarkable decrease in the median survival timings of the patients. Therefore, the high-level expression may work as a prognostic indicator for severe disease, low patient survival, and insignificant tumour differentiation. As αvβ6 has a central role in the pathological course, its activity and expression have been extensively reviewed [13,28,33]. Various molecular mechanisms involving signalling cascade pathways contribute to the overall perfect functioning of the integrin αvβ6 (Figure 3).

This αvβ6 is generally accepted as a suppressor of the tumour process in healthy epithelial cells through its anti-proliferative actions by activating the TGF-β1 in these cells. A scientific study on ITGB6 knockout animals disclosed that they did not develop any symptoms of malignant or even benign tumours [30]. In the course of tumour development, together with genetic and epigenetic manipulations, the obscure balance between αvβ6 and TGF-β1 goes uncoupled, resulting in inoperative TGF-β1-mediated signalling pathways where TGF-β1 changes from tumour suppressor to tumour promoter [34]. Consequently, both αvβ6 and TGF-β1 are linked to the tumour process when present in excess amounts in the body. The disruption of the homeostasis maintained between them has been proposed to occur through various mechanisms. The well-studied explanation for the loss of the growth inhibitory action of TGF-β1 is the mutations that occur in growth regulatory genes, e.g., the c-MYC mutation has been most frequently identified in human cancers [29].

Most actions of TGF-β1 are elicited through the canonical SMAD/non-SMAD pathway, but in the case of mutation in any regulatory signalling molecule, TGF-β1 redirects its actions by adopting other transduction pathways [35]. In addition, the commonly occurring oncogenic activation of extracellular signal-regulated kinase (ERK) pathway/mitogen-activated protein kinase/Ras/epidermal growth factor (EGFR) may also cause attenuation of the SMAD pathway by intervening and inhibiting the phosphorylation of the SMAD2/3 linker region [36]. Generally, the SMAD3 linker region undergoes phosphorylation in response to cancer and replaces the aberrant actions of TGF-β1 [35]. It is also possible that the signalling pathways independent of TGF-β1 control the ITGB6 expressions in cancerous cells. As an example, STAT3, which is a transcription factor constantly activated during different cancers, regulates ITGB6 as well as several other genes that perform a role in cell proliferation, invasion, survival, and angiogenesis. STAT3 may also cause an attenuation of TGFβ1-mediated growth inhibition and transcription inside epithelial cells by directly interacting with the SMAD3 pathway, and as a result, an anti-suppressing effect appears [37]. Another factor, Ets-1, is found to be up-regulated in the tumour process, wherein it boosts the ITGB6 expression and cell invasions [24,38]. There is an alternative mechanism of TGF-β1 activation other than by the integrin αvβ6: TGF-β1–uPA interaction in which TGF-β1 promotes the expression of urokinase plasminogen activator (uPA), which in turn activates the latent TGF-β1 complex by plasminogen activation [39].

The key to TGF-β1 transformation from tumour suppressor into promoter in various malignant cancers lies in the robust activation of the Ras-ERK signalling pathway [34]. Eventually, TGF-β1 accompanied by other growth factor signalling cascades can stimulate Epithelial-Mesenchymal Transition (EMT) in carcinoma cells. This transition permits them to relax their organization, producing a more subtle mesenchymal phenotype which facilitates the invasion. EMT induces Ets-1, which then activates the ITGB6 transcription, αvβ6 expression, and latent TGF-β1 complex [24]. The role of integrin αvβ6 goes further than the activation of TGF-β1, as its expression alone can impact the carcinoma cell phenotype. In a study on CRC cells, the overexpressed αvβ6 along with 708 expressed proteins including potential 54 cancer biomarkers, was found to increase cell proliferation and invasion, determining the deep effect of this protein receptor on carcinogenic cell biology. Notably, the defined role of EMT includes invasiveness into cultured epithelia cells, as demonstrated by experimental 3D invasion models as well as animal tumour models. Thus, the necessity of EMT for in vivo metastasis poses technical challenges, and this area attracts scientific investigations [40].

Pro-invasive proteins are clustered by the active functioning of αvβ6 at the tumour-infiltrating edge. These proteins gather uPA and TGF-β type II receptors and move the proteolytic signalling activities of PA cascade and TGF-β1 towards the tumour front, invading their surroundings [24,38]. The central role of integrin αvβ6 in tumour invasion is mainly performed by the intracellular cytoplasmic tail of the β6 subunit. The β6 subunit is directly targeted by ERK2 during the ERK activation in cancerous cells [9,17]. If this binding is prevented, cancer growth will be inhibited. In a similar way, Matrix metalloproteinases MMP-3 and MMP-9 that undergo induction and then activate the cancerous invasion are also dependent on the β6 cytoplasmic tail. Their induction can be abolished by knocking-down Ets-1 or by inhibiting the Ras/ERK pathway. MMP-2 induction is also carried out through Ets-1-dependent β6-mediated mechanisms. Another molecule, psoriasin (S100A7), may also participate in the αvβ6-based carcinoma invasions by binding to the integrin β6 cytoplasmic tail [13].

As a receptor to the extracellular matrix (ECM), αvβ6 also proceeds the cancerous cell migration on the ECM. High fibronectin expressions are linked with advanced stage tumours [41]. In a research study, in vitro experiments were performed on breast cancer cells where they found the integrin αvβ6 promoting the cell migration and invasion localized on MMP-9 degraded fibronectins, also assisting in the metastasis of cancer cells through fibronectin-mediated extravasations. Tenascin-C is another ligand that has been observed with integrin αvβ6 at the invasive tumour front in cancers, and it offers poor clinical outcomes in the case of severe malignant conditions [13].

A study has shown the possible migration of αvβ6 between the prostate cancerous cells through cell-derived vesicles, which are further taken up by the recipient cells deficient in integrin αvβ6. This protein becomes localized on the cell surfaces. Such paracrine-fashioned transportation of the αvβ6-affiliated malignancies makes the metastatic behaviour of tumour cells feasible to the neighbour cell [13]. Table 1 summarizes the mechanistic implications and role of αvβ6 upon expression in different cancers.

## 5. Integrin αvβ6 as a Target for Imaging and Therapy

As the integrin αvβ6 expresses with undetectable levels in normal healthy tissues and with high prevalence in invasive tumour cells, it may act as a potential target for tumour imaging and clinical investigations, leading to early diagnostics and therapeutics [28,60]. It also serves as a prognostic marker for tracking pre-invasive tumour progression (e.g., in ductal carcinoma) and monitoring the treatment. Due to its vital role in tumorigenesis, it is a promising target for different treatment approaches. Effective targeting compounds/agents are being developed to perform tumour imaging and delivering therapeutic drugs to cancerous cells [13]. Table 2 summarizes the therapeutic approaches in the targeting of αvβ6 and lists the drugs that are currently explored in different stages of clinical development (retrieved from Citeline Informa Database—Pharmaprojects^®^|Citeline, Drug Development Database|Pharma R&D|Pharmaprojects (informa.com) (https://pharmaintelligence.informa.com/products-and-services/clinical-planning/pharmaprojects) (accessed on 10 October 2022)).

Numerous potential anticancer strategies based on integrin αvβ6 have already been developed and some of them are described here. Immunoliposomes are the vectors used for intracellular targeted drug delivery. In the first step, antibodies (monoclonal or polyclonal) are attached to the liposomal surface and then, recognizing specific antigens, these bind onto tumour cell surfaces. The next step internalizes this drug vector and the targeted drug released in the intracellular compartments. Due to high specificity for drug release, immunoliposomes can deliver large quantities of drugs into cancer cells with the least side effects. In a study on colon cancer, immunoliposomes targeting ITGB6 were developed by combining ITGB6 monoclonal antibodies with PEGylated liposomes. These liposomes successfully delivered the antibodies into ITGB6-positive tumour cells and provided an efficient strategic approach for targeted drug delivery [28]. Two selective, potent monoclonal antibodies 6.8G6 and 6.3G9 were generated, which have high affinity for the human and mouse integrin αvβ6. These block αvβ6 for TGF-β LAP and prevent its activation. These can also block αvβ6-fibronectin interaction, thus, blocking the integrin αvβ6-mediated functions. Neither antibody could bind αvβ6 simultaneously, indicating that they may possess specificities for the overlapping epitopes [61,62,63]. Zhao-yang et al. used a mouse-anti-human β6 function-blocking monoclonal antibody 10D5 to block integrin αvβ6 and found the expression of the phosphorylated extracellular signal-related kinase (P-ERK) significantly decreased. ERK-activation has shown anti-apoptosis effects in colon cancer cells. Integrin αvβ6 plays a vital role in apoptosis-inhibition during cancer and has a significant impact on the mitochondrial pathway [64]. In another study on 10D5-mediated integrin αvβ6-blocking, tumour cell migration, as well as ERK activation, were markedly inhibited, both of which are usually induced by VEGF. Exposing the cells to β6 function-blocking 10D5 antibody inhibits 75% cell migration, in comparison with that observed in the healthy control group [32]. During a rationalized approach, a humanized scFv (single chain Fv) antibody, B6-3, was synthesized with therapeutic potential, causing αvβ6-blockage, which inhibits tumour cell invasion. This antibody can be used for the specific delivery and internalization of the cargo, e.g., toxins specific to αvβ6-expressing cancers [65].

A potential αvβ6-based therapy is human antibody 264RAD, which is highly specific for the integrin αvβ6, blocking the binding of the receptors with their cognate ligands and thereby inhibiting the αvβ6 biological functions. Hence, this therapy remarkably suppresses tumour proliferation and survival [34,66]. In another research study, 264RAD antibody has also demonstrated its effectiveness in reducing tumour growth as well as enhancing the therapeutic effect in combination with trastuzumab in breast cancer [28]. The novel recombinant B6.3 diabody antibody binds specifically to αvβ6 in vitro and targets the specific αvβ6-expressing tumours in vivo. This diabody was engineered with a C-terminal hexahistidine tag (His tag), which was purified by immobilized metal affinity chromatography (IMAC) from Pichia pastoris. The ligand-mimicking diabody elicited αvβ6 internalization upon binding, and effectively blocked αvβ6-dependent adhesion and migration to fibronectin and LAP. It suppressed smad2/3 nuclear translocation upon treatment with latent TGFβ1. The function-blocking ability and target-specificity make the B6.3 diabody either a potent imaging agent or a potential source for the generation of therapeutics through a chemical coupling of small cytotoxic molecules or the addition of toxic agents [67]. These studies proved that αvβ6 is a potential and promising anticancer therapeutic target for clinical practices. In a very recent study on pancreatic cancer, an FMDV-peptide drug conjugate SG3299 was developed by combining αvβ6-specific 20mer peptide from the VP1 coat protein of foot-and-mouth-disease virus (FMDV) with the DNA-binding pyrrolobenzodiazepine (PBD)-based payload SG3249 (tesirine). This resultant conjugate exhibited αvβ6-specificity and -selectivity in vivo and in vitro, as well as showing promising outcomes in selectively eliminating the αvβ6-positive tumour, providing a novel molecular therapy for the patient with pancreatic cancer [68].

Another example is the use of anti-αvβ6 antibodies in combinational therapy in αvβ6-expressing colorectal cancer. The antibody-mediated inhibition of integrin αvβ6 initiated a potent cytotoxic T-cell response and overcame resistance to programmed cell death protein 1 (PD-1)-targeting therapy, provoking a substantial increase in anti-PD-1 treatment efficacy [69,70,71].

Considering the differential expression, αvβ6 integrin is believed to be a novel target for cancer imaging. A series of experimental methods have been devised in various studies, most of which comprise molecular imaging, e.g., single-photon emission computed tomography (SPECT) and positron emission tomography (PET) [72]. As a preclinical development in a recent study, the [^18^F] αvβ6-binding peptide was used for αvβ6-based imaging through first-in-human (non-invasive) PET imaging in metastatic carcinoma, which demonstrated high clinical impact for a variety of malignancies [73]. The A20FMDV2 (NAVPNLRGDLQVLAQKVART) peptide has been derived from the foot and mouth virus, possesses high selectivity and affinity for the αvβ6 integrin’s RGD site, and can be employed as a radiotracer after being radiolabelled with the radionuclide. A scientific group radiolabelled this peptide with 4-[^18^F] fluorobenzoic acid and used microPET for cancer imaging. They achieved targeted cancer images of αvβ6-positive tumour cells because of the rapid uptake of radiolabelled peptide and then the selective retention of radioactivity in αvβ6-positive tumour cells. In other scientific studies, conjugated polyethylene glycol (PEG); small, monodispersed polymers; and new radiotracers: [^18^F] FBA-PEG28-A20FMDV2 and [^18^F] FBA-(PEG28)2-A20FMDV2 were generated for the improvement of pharmacokinetics. The former radiotracer exhibited improved PET-imaging even better than [^18^F]-FDG ([^18^F] fluoro-2-deoxy-D-glucose) during the BxPC-3 tumour imaging. A later study also showed a promising pharmacokinetic profile as well as tumour uptake for the [^18^F] FBA-PEG28-A20FMDV2 tracer. In another study, four chelators were selected and conjugated onto PEG28-A20FMDV2 for the radiolabelling of A20FMDV2 with ^64^Cu, but no best-suitable candidate was found [28].

The [^64^Cu] NOTA-αvβ6 cys-diabody was used for imaging αvβ6-specific tumours in vivo by a knockout mouse model bearing human melanoma xenografts, and the αvβ6-positive tumour cells were successfully visualised through PET imaging [74]. The A20FMDV2 peptide was also labelled with Indium-111, which showed good tumour retention and excellent discriminating tumour images by the use of biodistribution and SPECT-imaging. Several other radiotracers have also been well-studied, such as ^99^mTc-6-hydrazinonicotinyl (tricine) (TPPTS)-HK (an integrin αvβ6-targeting peptide); 125I/131I-labeled HBP-1; and ^99^mTc-SAAC-S02 (cystine knot peptide, S02, with a single amino acid chelate (SAAC) and labelled with [^99^mTc(H_2_O)_3_(CO)_3_]+). Hal and co-workers identified a novel integrin αvβ6 RGD-mimetic radioligand with high affinity and selectivity: (S)-3-(3-(3,5-dimethyl-1H-pyr-azol-1-yl) phenyl)-4-((R)-3-(2-(5,6,7,8-tetrahydro-1,8-naphthyridin-2-yl)ethyl)pyrrolidin-1yl) butanoic acid radiolabelled [^3^H]. However, further investigation is required to explore its pharmacokinetic profile and imaging effect [28].

The integrin family of proteins, αvβ6, have a strong potential to be considered one of the most promising therapeutic targets for cancer. αvβ6 is upregulated on multiple cancers, providing a window of opportunity for the selective delivery of effective therapies for life-threatening conditions. Even though the evidence that the expression of integrin αvβ6 appeared to be a prognostic marker in cancer for over two decades, there have been relatively few clinical trials that have investigated targeting this integrin.

An example is the Phase I/II trial [NCT01008475] described by Merck in 2014. In this study, a monoclonal antibody targeting αv integrins, in combination with irinotecan and cetuximab in K-ras wild-type metastatic colorectal cancer patients was used. Although the primary point of this trial was not met, the analysis of retrospective subgroup results suggested that patients with a high expression of αvβ6 integrin responded well to the addition of Abituzumab to the standard of care, with improved progression-free survival (PFS) and overall survival (OS) compared with those tumours of low αvβ6 expression [75]. Currently, there are two active clinical trials in which [^18^F]-αvβ6-BP is used to visualize the disease progression in different types of cancers: lung, breast, brain, colorectal, and a single trial with drug PLN-74809 currently developed by Pliant Therapeutics; data are summarized in Table 3. (Search of αvβ6-List Results-ClinicalTrials.gov, accessed: October 2022 [76]) (see also Table 2).

Undoubtedly, in order to use the integrin αvβ6 as a target for tumour treatment, a thoughtful strategy is first required. Starting from a biological molecule, an antibody, or a smaller peptide or a conjugate thereof, all expected and unexpected side effects should be considered in the assumption of the detailed mechanism of action.

## 6. Conclusions

Failure of integrin αvβ6-based signalling mechanisms may lead to cancer and metastasis resulting from irregular cell division, adhesion, and migration. The integrin αvβ6 can serve as a prognostic indicator due to up-regulation in various carcinomas and expression correlation with the patient’s survival. Vast research investigations prove that this protein plays a significant role in tumorigenesis because it modulates the malignant behaviours of carcinogenic cells. The αvβ6-mediated tumour progression involves several oncogenic signalling pathways. Both the expression and vital role of αvβ6 in the process of tumorigenesis and tumour progression have paved the way toward αvβ6-targeted studies for tumour imaging and therapeutic interventions, all of which have made αvβ6 a promising target for future genetic and clinical investigations, as well as appropriate treatment options. Although there is a large amount of research describing promising results to support the thesis that αvβ6 integrin could be an appropriate target for various cancer treatments, there are still sparse ongoing clinical trials and an unmet need from diseased patient perspectives.

## Figures and Tables

**Figure 1 ijms-23-12346-f001:**
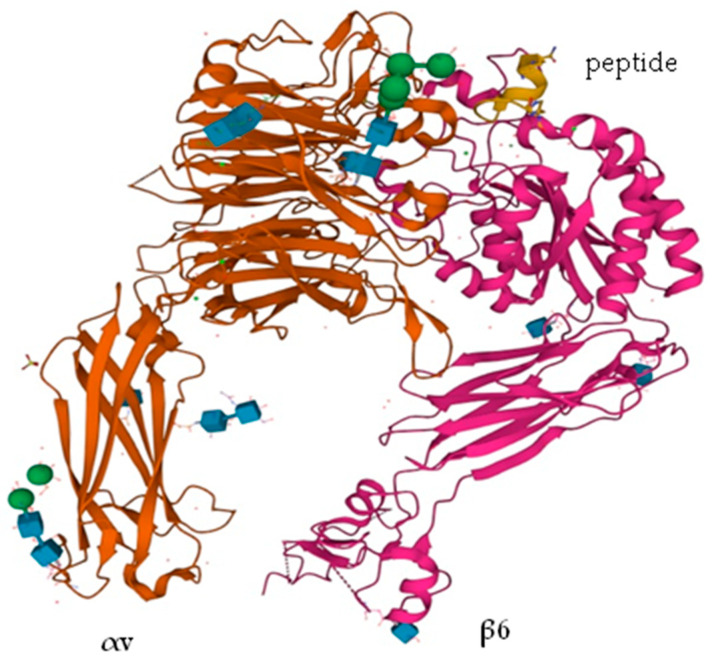
Crystal structure of αVβ6 headpiece with peptide (yellow) PDB ID: 4UM9 [7].

**Figure 2 ijms-23-12346-f002:**
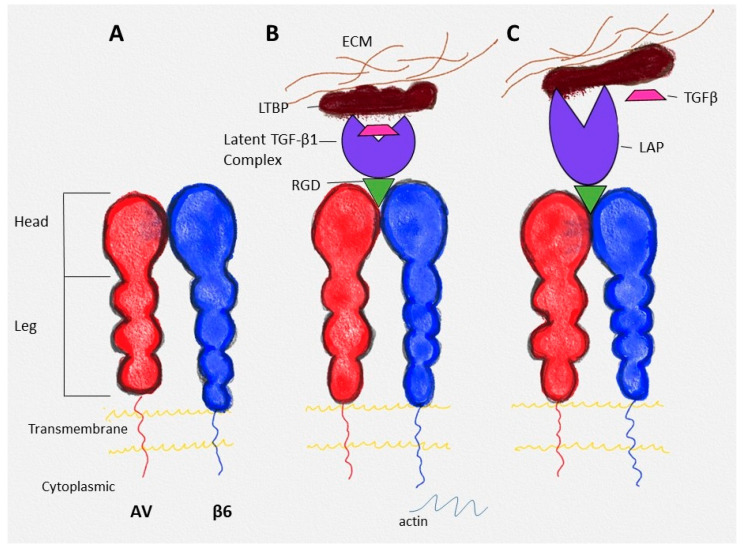
(**A**) Structure of integrin αvβ6; (**B**) binding of latent TGF-β complex with αvβ6 (the complex is bound to the latent TGF-β1 binding protein—LTBP); (**C**) activation of TGF-β (modified) [13]. Reprinted/adapted with permission from Ref. [13]. 2022, Elsevier.

**Figure 3 ijms-23-12346-f003:**
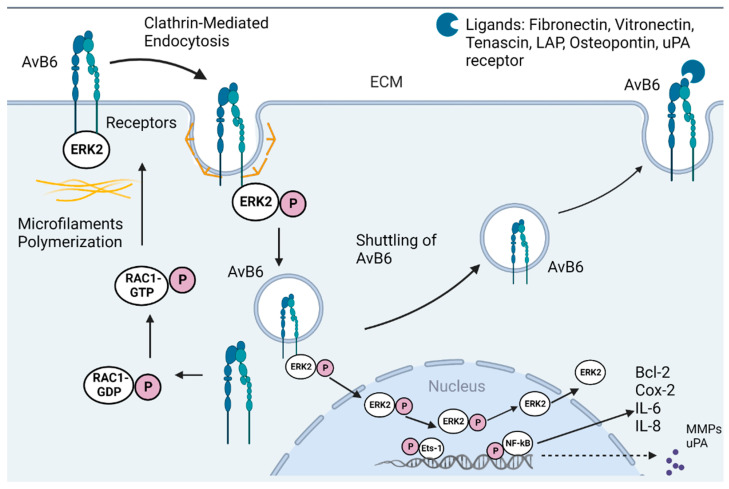
Schematic representation of the role of integrin αvβ6 in cancer. Clathrin-dependent endocytosis of integrin αvβ6 activates ERK signalling cascade. Activated ERK results in phosphorylation of Ets-1 and NF-kB pathway. This results in transcription and translation of MMPs, Bcl-2, uPA, and IL-6. The activated Ets-1, upon localizing in the nucleus, results in integrin αvβ6 transcription and eventual recycling within the tumour environment. This results in a positive feedback look and promotes cancer tumour proliferation and disease progression (modified [28]). Reprinted/adapted with permission from Ref. [28]. 2022, Elsevier.

**Table 1 ijms-23-12346-t001:** Mechanistic implications and role of αvβ6 upon expression in different cancers.

Cancer	Functions
Endometrial	Promotes invasiveness [41]
Basal Cell	Promotes invasiveness [42]
Liver	Poor prognosis [43]
Colon Cancer	Involved in cell adhesion and migration, activation of TGF-B, regulation of extracellular proteases, poor prognosis marker and implications in cancer metastasis to liver [43,44,45,46,47,48]
Gastric	Prognostic marker [49]
Cervical Squamous	Unfavourable marker [50]
Oral SCC	Increased expression in invasive stage of the cancer, regulation of migration and adhesion of cancer cells [51,52,53]
Pancreatic	Enhanced expression, regulation of tumour angiogenesis [54,55]
Breast	EMT induction and invasiveness [56,57]
Ovary	Enhanced expression leads to promotion of metastasis [58,59]

**Table 2 ijms-23-12346-t002:** Summary of drugs targeting αvβ6 in various stages of clinical development. Retrieved data from Citeline’s Pharmaprojects database: Pharmaprojects^®^|Citeline. (Pharmaprojects®. Citeline. 2022. Available online: Drug Development Database|Pharma R&D|Pharmaprojects (informa.com) (https://pharmaintelligence.informa.com/products-and-services/clinical-planning/pharmaprojects) (accessed on 10 October 2022).

Company	Target	Molecule ID/Name	Molecule Type	Drug Disease	Highest Status Reached	Summary
AstraZeneca	integrin subunit alpha V integrin subunit beta 6	264RAD	Antibody	Cancer, pancreatic	Preclinical, no active development reported.	264RAD is a human therapeutic antibody targeting alphavβ6 integrin, which was under development by AstraZeneca for the treatment of pancreatic cancer.
Aura Biosciences Cancer Research Technology	integrin subunit alpha V integrin subunit beta 6	[^19^F] FAB-A20FMDV2; A20FMDV2;	Synthetic peptide	Cancer, pancreatic diagnosis, cancer	Preclinical, no active development reported.	A20FMDV2 is a lead integrin alphaVβ6 binding peptides, which was under development by Cancer Research Technology (CRT) for use in targeted delivery of anticancer compounds and for cancer imaging.
Merck KGaA	integrin subunit alpha V integrin subunit beta 1 integrin subunit beta 3 integrin subunit beta 5 integrin subunit beta 6 integrin subunit beta 8	Abituzumab	Antibody	Cancer, solid	Phase II, no active development reported.	Abituzumab is an investigational integrin inhibiting monoclonal antibody with activity against avβ1, 3, 5, 6, and 8 integrin heterodimers, which was under development by Merck KGaA for the treatment of colorectal cancer. It was previously developed for systemic sclerosis.
Venn Therapeutics	integrin subunit alpha V integrin subunit beta 6 integrin subunit beta 8 transmembrane protein 173	Ad-VCA0848; AdVCA0848; VTX 002;	Antibody	Cancer, unspecified. Chronic obstructive pulmonary disease, idiopathic pulmonary fibrosis	Pre-clinical, no active development reported.	Ad-VCA-0848 (VTX-002) is a novel, immunotherapeutic off-the-shelf humanized monoclonal antibody STING agonist which recombined a potent diguanylate cyclase gene, VCA-0848, into a nonreplicating adenovirus serotype 5 targeting both alphavbeta8 and alphavbeta6, which was under development by Venn Therapeutics for the treatment of pulmonary fibrosis and COPD. It was previously under development for the treatment of cancer.
GlaxoSmithKline	integrin subunit beta 6	alpha v integrin antagonists, GlaxoSmithKline; avβ6 antagonists; avβ6 integrin anatagonists; integrin avβ6 anatagonists	Small molecule	Idiopathic pulmonary fibrosis	Preclinical, no active development reported.	GlaxoSmithKline was developing alpha v beta 6 integrin antagonists for the treatment of pulmonary fibrosis.
DiCE Therapeutics	integrin subunit alpha V integrin subunit beta 1 integrin subunit beta 6	alphaVbeta1/alphaVbeta6 integrin antagonist, DiCE Therapeutics; aVβ1/aVβ6 integrin antagonist, DiCE Therapeutics	Small molecule	Idiopathic pulmonary fibrosis	Preclinical, development active	DiCE Therapeutics (previously Dice Molecules) is developing an orally available alphaVbeta1/alphaVbeta6 (aVβ1/aVβ6) integrin antagonist using its DELSCAPE platform, for the treatment of idiopathic pulmonary fibrosis.
Seagen	integrin subunit alpha V integrin subunit beta 6	anti-alpha-V-beta6 ADC, Seattle; anti-avβ6 ADC, Seattle	Small molecule	Cancer, unspecified	Preclinical, no active development reported.	Seattle was developing a MAb against Integrin alpha-v-beta6, using its antibody–drug conjugate (ADC) technology, for the treatment of lung, pancreatic, and head and neck cancers.
Cancer Research Technology Cancer Research Technology	integrin subunit alpha V integrin subunit beta 6	anti-alphavβ6 antibodies, CRT; anti-integrin avss6 antibodies, CRT	Antibody	Cancer, unspecified; fibrosis, unspecified	Preclinical, no active development reported.	Cancer Research Technology (CRT) was developing anti-alphavβ6 antibodies for the treatment of cancer and fibrotic diseases.
Biogen	integrin subunit alpha V integrin subunit beta 6	alphav beta 6 integrin inhibitor, Biogen Idec; BG 00011; BG-00011; BG00011; STX-100; STX100	Antibody	Cancer, unspecified; pulmonary fibrosis, nephropathy	Phase II clinical trials, no active development reported.	Biogen Idec (Stromedix) has discontinued development of BG-00011 (STX-100), a humanized mAb targeting the alpha v beta 6 integrin for the treatment of idiopathic pulmonary fibrosis (IPF), due to safety concerns.
Panorama Research Corbus Pharmaceuticals	integrin subunit alpha V integrin subunit beta 6 integrin subunit beta 8	CRB 602; CRB-602; CRB602	Antibody	Cancer, unspecified; fibrosis, unspecified	Preclinical, development active	CRB-602 is an anti-alphavbeta6 and alphavbeta8 (avβ6/8) monoclonal antibody under development by Panorama Research in collaboration with Corbus Pharmaceuticals for the treatment of fibrosis and cancer.
GlaxoSmithKline	integrin subunit alpha V integrin subunit beta 6	alpha V beta 6 integrin antagonists, GlaxoSmithKline; GSK 3008348; GSK-3008348; GSK3008348	Small molecule	Idiopathic pulmonary fibrosis	Phase I clinical trial, no active development reported.	GlaxoSmithKline has discontinued development of GSK-3008348, an alphaVbeta 6 integrin antagonist for the treatment of idiopathic pulmonary fibrosis, due to a lack of confidence in developability and portfolio considerations following an interim analysis of Phase Ib data. It is an inhaled formulation.
Indalo Therapeutics	integrin subunit alpha V integrin subunit beta 1 integrin subunit beta 3 integrin subunit beta 6	Fibrotic disease therapy, Indalo Therapeutics; IDL 2965; IDL-2965; IDL2965	Small molecule	Chronic renal failure, idiopathic pulmonary fibrosis, non-alcoholic steatohepatitis	Phase I clinical trial, no active development reported.	IDL-2965 is an oral, selective RGD-binding integrin (alphaVbeta1, alphaVbeta3 and alphaVbeta6) antagonist, which was under development by Indalo Therapeutics for the treatment of idiopathic pulmonary fibrosis and non-alcoholic steatohepatitis. RGD-binding integrin inhibits the activation of TGF-β as well as the ability of stiff extracellular matrix to promote fibroblast migration and survival.
Centocor	integrin subunit alpha V integrin subunit beta 3 integrin subunit beta 5 integrin subunit beta 6	BGB-101; BGB101; CNTO 95; CNTO-095; CNTO-95; intetumumab	Antibody	Cancer, solid	Phase II clinical trials, no active development reported.	Intetumumab is a high-affinity fully-human MAb, which was under development by Centocor (Johnson & Johnson (J&J)) for the treatment of cancer. It binds and inhibits integrins alphavβ1, alphavβ3, alphavβ5, alphavβ6 and alphavβ8, and has antiangiogenic activity It was generated using Medarex’s (now Bristol-Myers Squibb’s (BMS)) UltiMAb technology.
AbbVie/Morphic Therapeutic	integrin subunit alpha V integrin subunit beta 6	alphaVbeta6 small molecule inhibitor, Morphic Therapeutic; integrin therapies, Morphic Therapeutics; MORF 627; MORF beta6; MORF-627; MORF-beta6; MORF627; MORFbeta6	Small molecule	Non-alcoholic steatohepatitis, Primary sclerosing cholangitis	Preclinical, no active development reported.	AbbVie has discontinued development of MORF-627 (MORF-beta6), an oral selective alphaVbeta6 small-molecule inhibitor licensed from Morphic Therapeutic, for the treatment of primary sclerosing cholangitis due to safety concerns. It was previously under development for non-alcoholic steatohepatitis.
Morphic Therapeutic/AbbVie	integrin subunit alpha V integrin subunit beta 6	MORF 720; MORF-720; MORF720	Small molecule	Fibrosis, pulmonary, idiopathic	Preclinical, no active development reported.	AbbVie has discontinued development of MORF-720, an oral selective alphaVbeta6 small-molecule inhibitor, licensed from Morphic Therapeutic using its MlnT platform, for the treatment of idiopathic pulmonary fibrosis.
Pliant Therapeutics	integrin subunit alpha V integrin subunit beta 1 integrin subunit beta 6	avβ6/avβ1 dual integrin inhibitor, Pliant Therapeutics; PLN 1561; PLN-1561; PLN1561	Small molecule	Primary sclerosing cholangitis	Preclinical, no active development reported.	PLN-1561 is an avβ6/aVβ1 dual integrin inhibitor, which was under development by Pliant Therapeutics for the treatment of primary sclerosing cholangitis.
Pliant Therapeutics	integrin subunit alpha V integrin subunit beta 6	aVβ6 inhibitor, Pliant Therapeutics; PLN 1705; PLN-1705; PLN1705	Small molecule	Primary sclerosing cholangitis	Preclinical, no active development reported.	PLN-1705 is an aVβ6 selective inhibitor, which was under development by Pliant Therapeutics for the treatment of primary sclerosing cholangitis.
Pliant Therapeutics	integrin subunit alpha V integrin subunit beta 1 integrin subunit beta 6	avβ1/avβ6 dual integrin inhibitor, Pliant; avβ1/avβ6 integrin inhibitor, Pliant; PLN 74809; PLN-74809; PLN74809	Small Molecule	Idiopathic pulmonary fibrosis, primary sclerosing cholangitis, COVID-19 complications, acute respiratory distress syndrome	Phase II clinical trial, active development reported.	PLN-74809 is an oral, small-molecule avbeta1/avbeta 6 dual integrin inhibitor under development by Pliant Therapeutics for the treatment of idiopathic pulmonary fibrosis and primary sclerosing cholangitis. The avβ1/avβ6 integrin blocks the activation of TGF-β in a tissue-specific manner, preventing the growth of fibrotic tissue within the lung. It was previously under development for acute respiratory distress syndrome (ARDS) associated with COVID-19.
OcuTerra Therapeutics	integrin subunit alpha V integrin subunit beta 3 integrin subunit beta 6 integrin subunit beta 8	OTT 166; OTT-166; OTT166; SF 0166; SF-0166; SF0166	Small molecule	Retinopathy, diabetic macular degeneration, age-related, wet oedema, macular, diabetic	Phase II clinical trial, development active.	SF-0166 is a fluorinated, selective small-molecule inhibitor of integrin alphaVbeta3, alphaVbeta6 and alphaVbeta8, which is under development by SciFluor Life Sciences (now OcuTerra Therapeutics) for the treatment of diabetic retinopathy. It was previously in development for retinal disease, including wet age-related macular degeneration (AMD) and diabetic macular oedema (DME). It is administered topically to the eye.
Seagen	integrin subunit beta 6 secretory leukocyte peptidase inhibitor	SGN B6A; SGN-B6A; SGNB6A	Small molecule	Cancer, solid, unspecified	Phase I clinical trial, development active.	SGN-B6A is an antibody-drug conjugate (ADC) targeting integrin beta-6, which is under development by Seagen (formerly Seattle Genetics) for the treatment of solid tumours.
Ligand Pharmaceuticals	integrin subunit alpha 5 integrin subunit alpha V integrin subunit beta 1 integrin subunit beta 3 integrin subunit beta 5 integrin subunit beta 6	XC 201; XC-201; XC201	Recombinant peptide	Cancer, unspecified	Preclinical, active development reported.	XC-201 is a peptide-Fc fusion (mAb-lite) targeting avβ1, avβ3, avβ5, avβ6, and a5β1 integrins, which was under development by xCella Biosciences for the treatment of cancer. A unique engineered integrin-binding peptide is created, and it is genetically fused to an antibody Fc domain. It is a universal tumour targeting agent that binds a broad panel of integrins (av and a5) expressed at high levels on tumours.

**Table 3 ijms-23-12346-t003:** List of active clinical trials with AvB6 in therapeutic settings and imaging. Different stages of clinical phase development are indicated. Retrieved from ClinicalTrials.gov, accessed on 10 October 2022 [76].

Title	Status	Conditions	Interventions	Locations	Phase	Reference
First-in-Human Positron Emission Tomography Study Using the 18FαVβ6-Binding-Peptide	Recruiting	Breast carcinoma|Colorectal carcinoma|Lung carcinoma|Metastatic malignant neoplasm in the breast|Metastatic malignant neoplasm in the colon|Metastatic malignant neoplasm in the lung|Metastatic malignant neoplasm in the rectum|Pancreatic carcinoma	Drug: 18F-αVβ6-BP	University of California Davis Comprehensive Cancer Center, Sacramento, California, United States	Early Phase 1	https://ClinicalTrials.gov/show/NCT03164486, accessed on 10 October 2022
Phase 2a Evaluation of PLN-74809 on αVβ6 Receptor Occupancy Using PET Imaging in Participants With IPF/	Recruiting	Idiopathic pulmonary fibrosis	Drug: PLN-74809|Radiation: Knottin tracer	Stanford Medical Center, Palo Alto, California, United States	Phase 2	https://ClinicalTrials.gov/show/NCT04072315, accessed on 10 October 2022
PET/CT Imaging in COVID-19 Patients	Enrolling by invitation	COVID-19|SARS-CoV-2 Infection	Drug: 18F-αVβ6-BP	University of California Davis, Sacramento, California, United States	Early Phase 1	https://ClinicalTrials.gov/show/NCT04376593, accessed on 10 October 2022

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
