# Peer review of "Integrin Alpha v Beta 6 (αvβ6) and Its Implications in Cancer Treatment"

_ijms, 2022, doi:10.3390/ijms232012346_

Round 1

Reviewer 1 Report

1. Research on mechanism should be strengthened.

2. Please add some figures and tables to make it easier for the reader to understand.

Author Response

We are very grateful for the comments and suggestions made by both reviewers. We have significantly improved the text, adding a table and a figure. We corrected all deficiencies pointing by the reviewer. I hope that the amendments that we have made will be satisfactory and that the revised manuscript will meet the International Journal of Molecular Sciences publication standards. Please find our response to criticisms and comments below. The inserted or corrected text is in yellow. I hope that we covered all the major revision points

  1. Research on mechanism should be strengthened.

The mechanistic implications and role of αvβ6 upon expression in different cancer has been added in the Table 1

  1. Please add some figures and tables to make it easier for the reader to understand.

The additional figure and a table have been added.

Reviewer 2 Report

Overall, the review "Integrin Alpha v Beta 6 (αvβ6) and its implications in cancer treatment" makes a good impression, a balanced set of cited articles was used. It is strange that the authors do not use the term theranostics in the fifth section of the review.

A small number of minor typos and formatting errors were detected during reading:
1) The abbreviation LBTP in Figure 1 is only deciphered in the text on the next page on line 114 and is not used further in the manuscript. Is it appropriate to use it.
2) In Figure 2 there is a typo - "vibronectin", probably vitronectin.
3) On line 239 there is an extra comma at the beginning of the sentence.
4) On line 242 the abbreviation IMAC is given without decoding.
5) On page 8 there are many formulas or names of compounds containing isotopes (fluorine-18, copper-64, indium-111, technetium-99, tritium), the corresponding numbers should be printed with an upper index.
After the indicated deficiencies have been corrected, the manuscript may be published without a second review.

Author Response

We are very grateful for the comments and suggestions made by both reviewers. We have significantly improved the text, adding a table and a figure. We corrected all deficiencies pointing by the reviewer. I hope that the amendments that we have made will be satisfactory and that the revised manuscript will meet the International Journal of Molecular Sciences publication standards. Please find our response to criticisms and comments below. The inserted or corrected text is in yellow. I hope that we covered all the major revision points.

I am looking forward to hearing from you.

Yours sincerely,

Ewa Brzozowska

Reviewer 1

Overall, the review "Integrin Alpha v Beta 6 (αvβ6) and its implications in cancer treatment" makes a good impression, a balanced set of cited articles was used. It is strange that the authors do not use the term theranostics in the fifth section of the review.

A small number of minor typos and formatting errors were detected during reading:
1) The abbreviation LBTP in Figure 1 is only deciphered in the text on the next page on line 114 and is not used further in the manuscript. Is it appropriate to use it.

The LBTP was decoded under the Figure 1 description.

2) In Figure 2 there is a typo - "vibronectin", probably vitronectin.

It has been corrected

3) On line 239 there is an extra comma at the beginning of the sentence.

It has been corrected

4) On line 242 the abbreviation IMAC is given without decoding.

It has been decoded

5) On page 8 there are many formulas or names of compounds containing isotopes (fluorine-18, copper-64, indium-111, technetium-99, tritium), the corresponding numbers should be printed with an upper index.

It has been corrected

After the indicated deficiencies have been corrected, the manuscript may be published without a second review

Round 2

Reviewer 1 Report

There are too few figures and tables. The manuscript is too short and not deep enough.

Author Response

Dear Reviewer,

We appreciate the reviewer’s valuable feedback. We agree with the reviewer’s opinion that including tables and figures reflecting the descriptive text in the manuscript will improve the readability of the paper and make it more fluid to develop a better understanding of the subject.

To address the reviewer’s comments, we have now included additional three tables and one additional figure as follows:

 Table 1:  Summarizes implications and mechanistic role of AvB6 in different cancers.

Table 2: Summary of active and non-active drug development programs and strategies targeting AvB6 in clinical settings. This table provides a helicopter view of all the reported efforts from the pharmaceutical industry from  clinical and commercial perspective, until October 2022.

Table 3: List of active clinical trials with AvB6 in therapeutics and imaging. Different stages of clinical phase development are also indicated until October 2022.

Figure1: We have now included a crystal structure of αVβ6 headpiece in complex with its peptide. We have also updated the text to reflect the binding and complex interplay of AvB6 and its peptide.

In addition, the text body is updated wherever necessary to refer to the tables and figures.

References are also corrected and updated.  

We believe that with these changes; we have addressed the reviewer’s comments and substantially improved the manuscript's quality.

We thank all the reviewers for the positive and constructive feedback to implement the changes and improve the quality of our work.

Best Regards,

Ewa Brzozowska

Round 3

Reviewer 1 Report

Accept